# Posttraumatic Growth and Spirituality in Mothers of Children with Pediatric Cancer

**DOI:** 10.3390/ijerph18062890

**Published:** 2021-03-12

**Authors:** Natalia Czyżowska, Magdalena Raszka, Alicja Kalus, Dorota Czyżowska

**Affiliations:** 1Institute of Psychology, Pedagogical University of Kraków, 30-084 Kraków, Poland; natalia.czyzowska@up.krakow.pl; 2Independent Researcher, 44-200 Rybnik, Poland; raszka.magdalena12@gmail.com; 3Institute of Psychology, Opole University, 45-052 Opole, Poland; 4Institute of Psychology, Jagiellonian University, 30-060 Kraków, Poland; d.czyzowska@uj.edu.pl

**Keywords:** pediatric cancer, mothers, posttraumatic growth, spirituality

## Abstract

A child’s cancer, as a life-threatening illness, is classified as a traumatic event both for the child him-/herself and for his/her relatives. Struggling with a traumatic experience can bring positive consequences for an individual, which is referred to as posttraumatic growth. The aim of this study was to explore the relationship between posttraumatic growth and spirituality understood as a personal resource in mothers of children with pediatric cancer. In total, 55 mothers whose children were in the phase of treatment and who had been staying with them in the hospital filled in a Posttraumatic Growth Inventory, Self-description Questionnaire of Spirituality, and the author’s short questionnaire on demographic variables and information on the child and his/her disease. A high level of posttraumatic development, especially in the area of life appreciation, was observed in the examined mothers. Spirituality was positively related to the emergence of positive change, in two particular components, ethical sensitivity and harmony. It seems that taking into account the area of spirituality when planning interventions and providing support in this group could foster coping with the situation and emergence of posttraumatic growth.

## 1. Introduction 

It is estimated that each year approximately 300,000 children around the world are diagnosed with cancer and it is the leading cause of death in children and adolescents [1]. The situation of a child’s serious illness, especially one that requires hospitalization, has grave consequences for the whole family and may result in the appearance of symptoms of anxiety, depression or posttraumatic stress [2,3,4]. Researchers note that there are certain specific cancer-related stressors that parents experience in domains such as daily/role functioning, cancer communication, and cancer caregiving [5] or a sense of horror and helplessness during diagnosis and treatment [6]. Parental cancer-related stress levels are higher in mothers than in fathers [7,8], but it seems to be related to the age of the child—the older the child, the smaller the difference in experienced stress levels between parents, so if possible, the level of distress in both parent should be monitored, given the high risk of the negative consequences following pediatric cancer diagnosis [9].

However, it is worth noting that there is an increasing number of reports proving that experiencing traumatic life events can also bring positive psychological consequences for the individual [10,11]. The very idea that pushing through serious adversities can lead to significant positive changes, i.e., the growth and development of an individual, is nothing new and its origins date back to ancient times; it was also noticed by significant 20^th^-century psychiatrists and psychotherapists [12]. However, the problem of posttraumatic growth (PTG) in mainstream psychology was introduced by Tedeschi and Calhoun [13,14], who defined it as “positive psychological change experienced as a result of the struggle with highly challenging life circumstances” [15]. Posttraumatic growth is a multidimensional construct that includes changes in various areas of an individual’s life, such as self-perception, relationships with other people, spirituality, and philosophy of life. It is associated with a greater appreciation of life in general and various aspects of everyday life, such as the smile of a child or the opportunity to spend time with someone close, as well as feeling grateful for what you have. In the domain of relationships, people feel more connected with others and experience greater closeness; they are also more understanding and empathetic then before. After dealing with a traumatic experience, people can also see themselves as stronger and more resourceful, capable of dealing with a variety of situations, while also being more open and ready to notice and take advantage of new opportunities [15,16]. In the area of spirituality, we can observe the occurrence of new religious or spiritual behaviors or greater involvement in religious practices. However, it should be noted that people who do not identify themselves with any religion can experience spiritual growth, e.g., through more reflection on fundamental existential questions [15,17]. Tedeschi and Calhoun [15] emphasize that post-traumatic growth is not the same as resilience because resilience, unlike posttraumatic growth, is not transformative. Some researchers believe, however, that posttraumatic growth should be understood as a specific form of resilience [18]. In our research, we wanted to capture the changes that occur in the respondents as a result of coping with a child’s disease, so we decided to focus on PTG. There are studies showing that both children and adolescents with cancer and their parents experience posttraumatic growth [19,20]. The factors that may favor posttraumatic growth include optimism [21], constructive coping strategies (i.e., positive reframing) [22] social support, and spirituality [23].

Over the past two decades, many definitions of spirituality have emerged [24,25] and although some of them largely equate spirituality with religiosity, it seems reasonable to treat these phenomena as separate constructs. The results of the research indicate that there are people who define themselves as only spiritual or only religious [26]. Sheridan defines spirituality as “the search for meaning, purpose, and connection with self, others, the universe, and ultimate reality, however one understands it, which may or may not be expressed through religious forms or institutions.” [27], p. 10. Spirituality can also be related to the feeling of being connected to something greater than ourselves, something that gives a sense of the holy [28], a search for the sacred [25]. The relationship between posttraumatic growth and spirituality can be considered in two ways. On the one hand, traumatic experiences can contribute to spiritual development; on the other hand, spirituality can be treated as an individual’s resource that helps cope with a difficult situation [24]. In our research, we will focus on spirituality as an individual resource. While there have been some reports of traumatic growth in parents of children with pediatric cancer in recent years, hardly any of them have focused on the relationship between posttraumatic growth and spirituality, which may play a significant role here, especially if we consider that it is possible to design interventions that take into account spirituality [29].

The aim of this study is to explore if mothers struggling with a child’s cancer experience posttraumatic growth and whether there is a relationship between spirituality and posttraumatic growth. Based on the previous research, we expect that in mothers of children who face a life-threatening disease, the most changes can be seen in the area of appreciating life [30]. We decided to focus our study on mothers because, as mentioned earlier, they experience higher levels of stress than fathers, and they usually play the role of the primary caregiver and accompany the child during hospitalization.

## 2. Materials and Methods

### 2.1. Participants

The study was conducted at the University Clinical Hospital in Wroclaw (Lower Silesia Province) and at the John Paul II Upper Silesian Child Health Centre in Katowice (Silesia Province) during December 2019 and January 2020 with the consent of the Hospital Management. Data were collected from 55 mothers of children with cancer (*M* = 36.16; *SD*= 6.19) (*M* = 6.43; *SD* = 4.61). Only mothers whose children were in the phase of treatment and who had been staying with them in the hospital for at least two weeks were invited to the study. The invitation to participate in the research was provided by psychologists working in the hospital. Participation in the study was voluntary, and the respondents had the right to resign at any time, which was previously informed. In order to ensure the greatest possible anonymity of the respondents, they gave their verbal consent to participate in the study. The study was conducted individually and was performed in accordance with the principles of the Declaration of Helsinki. Most of the participants came from cities (65.45%) and had at least a secondary education. The respondents differed from each other in terms of marital status but most of them were married (74.55%). The most common type of cancer in children was lymphoblastic leukemia (45.45%). Diagnoses included neuroblastomas (10.91%), acute lymphoblastic leukemia (7.27%), Ewing’s sarcoma (5.45%), and other (30.92%). The table below presents a detailed description of the participants (Table 1).

### 2.2. Measures

The study used the Posttraumatic Growth Inventory [31], the Self-description Questionnaire of Spirituality [32], and a short questionnaire on demographic variables and information on the child and his/her disease prepared by the authors of the study.

The Posttraumatic Growth Inventory (PTGI) is a tool for measuring positive changes resulting from experiencing a traumatic situation. The study used the Polish adaptation of this tool consisting of 21 items to which the participant responds on the scale from 0 to 5 by selecting one of six possible answers (0—“I did not experience this change”, 5—“I experienced this change to a very great degree”) [33]. The higher the result obtained, the higher the intensity of positive changes the person presents. The inventory measures four factors: *changes in self-perception*—a person, as a result of a traumatic event, begins to notice new possibilities in his/her life and feels stronger; *changes in relationships with others*—a person feels a strong relationship with other people, his/her level of empathy and altruism increases; *appreciation of life*—after experiencing a trauma, a person begins to change his/her current priorities, philosophy of life, and shows a tendency to appreciate every moment and day of their life; *spiritual changes*—a person comes to an increase in his/her religiosity and a deeper understanding of spiritual issues. The Polish version of the PTGI has satisfactory psychometric properties. Cronbach’s alpha index for all items was 0.93. The reliability coefficients for individual factors ranged from 0.63 to 0.87. The internal stability of the questionnaire was satisfactory and amounted to 0.74. In this study, Cronbach’s alpha was 0.90.

The Self-description Questionnaire of Spirituality is used to measure spirituality, understood as an individual’s personal resource [32]. The questionnaire consists of 20 items. The participants must respond to each statement by selecting answer on a 5-point Likert scale (5—“definitely yes” and 1—“definitely not”). The higher the overall score, the higher the level of spirituality felt by the individual. The Self-description Questionnaire of Spirituality consists of three subscales that measure three aspects of spirituality: *religious attitudes*—define religious experiences, their role in the daily life of an individual, and their influence on their moral choices; *ethical sensitivity*—refers to the level of ethical attitudes; *harmony*—defines the perception of the world as friendly as well as a sense of inner peace. The first two scales consist of seven items, while the last scale contains six items. Cronbach’s alpha index for all items was 0.88. The reliability coefficients for individual scales ranged from 0.77 to 0.94. In this study, Cronbach’s alpha was 0.91.

## 3. Results

The analysis of the results showed a relationship between spirituality and posttraumatic growth (r = 0.33). The overall score on the spirituality scale was positively correlated with changes in the area of relationships with others (r = 0.31) and with changes in the spiritual area (r = 0.46). The relationship between posttraumatic growth and the two subscales of spirituality, i.e. harmony (r = 0.35) and ethical sensitivity (r = 0.35), was also revealed. Furthermore, harmony was correlated with changes in self-perception (r = 0.34) and in relating to others (r = 0.30), while ethical sensitivity was correlated with changes in relating to others (r = 0.34). Detailed results are presented in Table 2.

The conducted analyses showed that the examined mothers experienced positive changes as a result of fighting a traumatic event such as a child’s cancer. Referring to the PTGI standards, 7.2% of mothers obtained low results (between 3 and 4 sten, i.e. Standard Ten scores), 51% obtained average results (between 5 and 6 sten), and 41.8% of mothers obtained high results (higher than or equal to 7 sten). The obtained results are shown in Table 3. Furthermore, three areas of posttraumatic development were checked in terms of which one caused the greatest changes. As predicted, mothers experienced the most changes in the area of appreciating life (*M* = 4.26). The result was calculated by dividing the obtained means by the number of statements for a given factor of posttraumatic growth (for the appreciation of the life factor, the number of items was three).

To check whether there were differences in the area of spirituality between the group of mothers with high posttraumatic growth (higher than or equal to 7 sten) and low/average posttraumatic growth (between 3 and 6 sten), a Student’s t-test was carried out (Table 4). Mothers with high posttraumatic growth presented higher results in the subscales harmony and ethical sensitivity. There were no differences between groups at the level of the overall score and the subscale religious attitudes.

## 4. Discussion

Our research shows a relationship between spirituality and posttraumatic growth in mothers of children with pediatric cancer. The higher the level of spirituality of the surveyed mothers, the more positive changes they experienced in coping with a difficult situation. This is consistent with the results obtained by other researchers [30,34]. Researchers note that spirituality has a profound impact on our beliefs and the meanings we give to different situations, so dealing with the relationship between spirituality and traumatic experience seems necessary fora fuller understanding of a person’s situation [35]. The observed relationships between harmony (subscale of spirituality) and changes in self-perception and relationships with other people allow us to assume that the search for balance, internal order, and consistency between our beliefs and actions taken may contribute to seeing oneself as a better person and positively influencing an individual’s interactions and the way he or she views interpersonal relationships in general. Ethical sensitivity also correlated with changes in relationships with other people, which allows us to suppose that reflexing over values and placing ethical values high in our own hierarchy of values may also contribute to a greater appreciation of interpersonal relationships. Knowledge of these relationships can be very useful for psychologists, clinicians, and interveners caring for families of children with pediatric cancer. It indicates the areas that should be included in the therapeutic interventions especially since previous research indicates that there is also a relationship between spirituality and coping [36,37].It is also known that spiritually-derived interventions bring positive results in work in other areas, e.g., in social work [38], so it seems that it would be worth taking a closer look at this topic in the context of working with families struggling with a child’s cancer.

As expected, mothers of children with pediatric cancer experienced posttraumatic growth. In total, 92.7% of the mothers had moderate or high levels of posttraumatic growth. The greatest number of positive changes was reported by mothers in the area of appreciating life, which is consistent with the results of studies involving parents of seriously ill children [30,39]. It is worth mentioning that the differences in spirituality between the group of mothers who experienced high posttraumatic growth and the group of mothers who experienced it at a low/moderate level concerned two subscales, harmony and ethical sensitivity, which may be another argument that these are areas that should be included in therapeutic interventions. It is worth emphasizing in the context of our research that posttraumatic growth was studied in mothers of children who were currently in hospital and undergoing treatment, while previous reports have focused more on parents of cancer survivors [6,19,40]. This means that mothers experienced positive changes while facing a difficult situation and were not sure whether the treatment of their children would be successful. It was also specific that only mothers who accompanied their children during hospitalization and were present with them in the ward participated in the study. On the one hand, they could therefore experience greater stress, not only observing disturbing symptoms in their children, but also in other patients on the ward; on the other hand, they could experience greater support from the medical staff and other women in the same situation.

There were no associations between posttraumatic growth and religious attitudes. The researchers point out that the detected relationships between religiosity and reaction to a traumatic event largely depend on the concept of religiosity adopted in the research and the type of trauma [41]. Perhaps if we measured positive religious coping, the centrality of religiosity or the frequency of religious practices in the lives of respondents, we would be able to discover such relationships. It is also worth noting that PTGI measures the change that has occurred in an individual in various areas of life. If the mothers in the study were already very religious, they may not have experienced a significant change in this area. Unfortunately, we do not have data on the level of mothers’ religiousness before their child’s illness.

### Limitations and Future Directions

It is known that both the level of stress [23,42] and social support [23,43] may favor the experience of posttraumatic growth, but these variables were not included in our study. The relationships observed in our research are moderate, which undoubtedly indicates a further need for research in this area. It is possible that the relationship between spirituality and posttraumatic growth may be mediated by other variables which were not included in our study. We are aware that our study used a correlation model of research, which means that we cannot draw unambiguous conclusions about the direction of the relationship between spirituality and posttraumatic growth. Our assumptions are based on the literature; however, it is very likely that spirituality and posttraumatic growth influence each other and it would be worthwhile to study these relationships using a different research model. Since spirituality cannot be equated with religiosity, it seems that it would be worthwhile to measure both of these constructs in future research, which was missing in this study, as it could contribute to a better understanding of the relationship between them and the experience of posttraumatic growth. Another limitation of the study is the fact that the group of respondents was rather small and homogeneous, which made it impossible, for example, to identify sociodemographic factors that could favor the emergence of posttraumatic growth, and these are factors that would be worth considering in future research. It also seems important to look at the variables related to the child (especially age) and disease (duration, type of treatment, number of hospitalizations) and prognoses, which may significantly affect how parents perceive this situation and how they cope with it.

## 5. Conclusions

Based on the data collected and analyzed by us, the following conclusions can be drawn:Mothers of children with pediatric cancer undergoing treatment experience posttraumatic growth, especially in the area of life appreciation.There is a positive association between posttraumatic growth and spirituality, in particular its two components, ethical sensitivity and harmony.The area of spirituality should be considered in planning interventions to support this group.

## Figures and Tables

**Table 1 ijerph-18-02890-t001:** Characteristics of the sample (*n* = 55 mothers).

	*M*	*SD*	Min	Max
Mother’s age	36.16	6.19	23.00	55.00
Child’s age *	6.43	4.61	0.60	17.00
	*N*		*%*	
Education				
Primary	0		0	
Vocational	11		27.27	
Secondary	15		20.00	
Higher	29		52.73	
Place of residence				
Village	19		34.55	
Town	36		65.45	
Marital status				
Single	4		7.27	
Nonmarital cohabitation	5		9.09	
Married	41		74.55	
Divorced	5		9.09	
Time since diagnosis **				
<1 year	40		72.73	
1–2 years	9		16.36	
2–5 years	5		9.09	
>5 years	1		1.82	
Child’s diagnosis (type of cancer, detailed data)				
Biphenotypic leukemia	1		1.82	
Lymphoblastic leukemia	25		45.45	
Lymphoma	1		1.82	
Glioblastoma	1		1.82	
Germinal tumor	1		1.82	
Brain tumor	1		1.82	
Wilm’s tumor	1		1.82	
Sarcoma	1		1.82	
Ewing’s sarcoma	3		5.45	
Synovial sarcoma	1		1.82	
Nasopharyngeal sarcoma	1		1.82	
Fetal sarcoma	1		1.82	
Soft tissue sarcoma	1		1.82	
Neuroblastoma	6		10.91	
Acute lymphoblastic leukemia	4		7.27	
Teratoma	1		1.82	
Adrenal cortex cancer	1		1.82	
Rhabdomyosarcoma ofshoulder	1		1.82	
Embryonal rhabdomyosarcoma	1		1.82	
Lymphatic system hyperplasia	1		1.82	
Anaplastic ependymoma	1		1.82	

* Although the age range of the children was wide, it is worth noting that only five children were between 14 and 17 years of age, and almost 80% of the children were younger than 10 years of age. ** Almost 86% of children were diagnosed with cancer within the last 18 months.

**Table 2 ijerph-18-02890-t002:** Results of the analysis of the rho-Spearman correlation between posttraumatic development and spirituality in mothers in the event of a child’s cancer (*n* = 55).

		1	2	3	4	5	6	7	8	9
Posttraumaticgrowth	1. Total result	-								
	2. Changes in self-perception	0.93 ***	-							
	3. Changes in relating to others	0.88 ***	0.75 ***	-						
	4. Appreciationforlife	0.45 ***	0.37 **	0.31 *	-					
	5. Spiritual changes	0.58 ***	0.41 **	0.40 ***	0.23	-				
Spirituality	6. Total result	0.33 *	0.2	0.31 *	0.08	0.46 **	-			
	7. Religiousattitudes	0.2	0.07	0.22	−0.04	0.48 ***	0.89 ***	-		
	8. Ethicalsensitivity	0.35 **	0.22	0.34 *	0.21	0.32 *	0.81 ***	0.58 ***	-	
	9. Harmony	0.35 **	0.34 *	0.30 *	0.07	0.25	0.68 ***	0.41 **	0.47 ***	-

** p* < 0.05; ** *p* < 0.01; *** *p* < 0.001.

**Table 3 ijerph-18-02890-t003:** Descriptive statistics of the studied variables (*n* = 55).

	*M*	*Me*	*SD*	*Min*	*Max*	*W* _S-W_	*p*	*SKE*	*K*
Posttraumatic growth									
Total result	75.69	76.00	14.63	33.00	100.00	0.97	0.289	−0.47	0.17
Changes in self-perception	30.44	31.00	7.89	8.00	43.00	0.95	0.035	−0.74	0.39
Changes in relations with others	26.04	27.00	5.55	11.00	35.00	0.96	0.050	−0.65	−0.07
Appreciation for life	12.80	13.00	1.82	7.00	15.00	0.91	0.001	−0.80	0.69
Spiritual changes	6.42	7.00	2.83	0.00	10.00	0.91	0.001	−0.79	−0.22
Spirituality									
Total result	73.15	76.00	13.21	38.00	97.00	0.97	0.170	−0.50	−0.31
Religious attitudes	25.20	28.00	7.87	7.00	35.00	0.92	0.001	−0.78	−0.25
Ethical sensitivity	28.55	29.00	4.13	20.00	35.00	0.96	0.059	−0.41	−0.65
Harmony	19.42	19.00	3.90	11.00	27.00	0.97	0.200	−0.30	−0.34

*W*_S-W_—Shapiro–Wilk test statistics; *SKE*—skewness; *K*—kurtosis.

**Table 4 ijerph-18-02890-t004:** Differences in spirituality.

	Mothers with High PTG		Mothers with Low/Average PTG		*t*	*p*
	*M*	*SD*	*M*	*SD*		
Total result	75.71	11.86	69.56	14.38	1.73	0.08
Religious attitudes	25.75	7.52	24.43	8.45	0.60	0.54
Ethical sensitivity	29.50	4.11	27.21	3.83	2.08	0.04
Harmony	20.5	3.36	17.91	4.16	2.54	0.01

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
