# Peer review of "Posttraumatic Growth and Spirituality in Mothers of Children with Pediatric Cancer"

_ijerph, 2021, doi:10.3390/ijerph18062890_

Round 1

Reviewer 1 Report

  1. The theme of the study is an important issue in clinical care.
  2. Childhood cancers have different types.  Different type of cancer has the different treatment course.  Are those the influencing factors in relating to the results of the study ?
  3. Please reorganize Table 2.   item no.6 should closer to no. 5 or no.7 ?

Author Response

Reviewer 1

  1. The theme of the study is an important issue in clinical care.
  2. Childhood cancers have different types.  Different type of cancer has the different treatment course.  Are those the influencing factors in relating to the results of the study ?

Response:  Thank you very much for your valuable comment. We agree that this may be an important factor, unfortunately, during our research, we did not collect such data, and therefore we indicated it as a limitation of our research (in section: 4.1 Limitations and Future Directions).

  1. Please reorganize Table 2.   item no.6 should closer to no. 5 or no.7 ?

Response: Item no. 6 should be closer to no. 7. The table was reorganised (p. 5)

Reviewer 2 Report

The manuscript approaches a highly relevant topic, with promising practical implications. Regarding the introduction, it would be important to review some bibliographic references on resilience (in particular family resilience) and other related constructs, since they are not obviously synonymous but are rather closed; the authors may consider to differentially defining those constructs in order to justify or enhance the relevance of choosing this one.

In the methods section, considering the wide range of ages and of the time since the diagnosis of cancer, it would be important to take this in consideration, maybe comparing groups or doing correlations on different subgroups. Also, half of the sample are mothers of children with lymphoblastic leukaemia: could the authors also include this in their analyses and/or add some information regarding the severity of prognosis. For sure, all of those variables may make the difference on the mothers’ posttraumatic growth.

In the introduction, authors refer that the stress of the mothers is higher than the one presented by the fathers when the children are younger. However, as  they grow older,  stress levels are more similar within both parents; since the age range is so high, authors should justify better their choice of the mothers in the current study and highlight this as a limitation in the discussion.

The correlations are significant, but they are also of small magnitude: it deserves more attention from the authors in the discussion; it is interesting that spirituality is correlated with posttraumatic growth, but it is also interesting that, apparently, both dimensions are not highly associated.

Author Response

Reviewer 2

Comments and Suggestions for Authors

The manuscript approaches a highly relevant topic, with promising practical implications. Regarding the introduction, it would be important to review some bibliographic references on resilience (in particular family resilience) and other related constructs, since they are not obviously synonymous but are rather closed; the authors may consider to differentially defining those constructs in order to justify or enhance the relevance of choosing this one.

Response: Thank you very much for your valuable comment. In the introduction, we have added a fragment in which we refer to the relationship between posttraumatic growth and resilience (p.2).

In the methods section, considering the wide range of ages and of the time since the diagnosis of cancer, it would be important to take this in consideration, maybe comparing groups or doing correlations on different subgroups. Also, half of the sample are mothers of children with lymphoblastic leukaemia: could the authors also include this in their analyses and/or add some information regarding the severity of prognosis. For sure, all of those variables may make the difference on the mothers’ posttraumatic growth.

Response: Thank you very much for your valuable comment . We have added information to the text  that although the age range of the children was wide, it is worth noting that only five children were between 14 and 17 years of age, and almost 80% of the children were younger than 10 years of age (p. 4). For this reason, we did not decide to analyze the subgroups (as there would be a very large disproportion in the number of respondents assigned to each subgroup). The situation is similar with the time since diagnosis (almost 86% of children were diagnosed with cancer within the last 18 months – we also have added this information to the text p.4). We agree that these factors (which were not included in our research) may play an important role, so we indicate this as a limitation (in section: 4.1 Limitations and Future Directions, p.7).

In the introduction, authors refer that the stress of the mothers is higher than the one presented by the fathers when the children are younger. However, as  they grow older,  stress levels are more similar within both parents; since the age range is so high, authors should justify better their choice of the mothers in the current study and highlight this as a limitation in the discussion.

Response: Thank you very much for your valuable comment. In our study, we decided to examine mothers also because they most often play the role of the main caregiver and accompany the child during hospitalization.

The correlations are significant, but they are also of small magnitude: it deserves more attention from the authors in the discussion; it is interesting that spirituality is correlated with posttraumatic growth, but it is also interesting that, apparently, both dimensions are not highly associated.

Response: Thank you very much for your valuable comment. We indicated in limitations that the strength of the observed relationships is moderate and that it can be influenced by other variables that were unfortunately not included in our research (4.1 Limitations and Future Directions, p. 7).

Reviewer 3 Report

This is interesting research- topic.  The chosen idea approach is relevant to the field of posttraumatic growth and spirituality.

Introduction was useful and good leading to the research question

Methods:

-  when was the study conducted, what is the time / Duration of the Study? And how were the mothers recruited? It is not somewhat clear!

- In the manuscript was not mentioned how long had been child cancer?

how long had the mothers experienced the children's cancer?, the discussion of this point will be an enrichment for the topic and the Temporal relationship between posttraumatic growth and spirituality

-              Is there ethical approval of the Study?

-              Are there exclusion criteria?

  • Measures the questionnaire tolls are explained in general good,

- A suggestion for the to the Title, so it may refer to the results like:

a positive association between posttraumatic growth and spirituality in mothers of children with pediatric cancer

Author Response

Reviewer 3

Comments and Suggestions for Authors

This is interesting research- topic.  The chosen idea approach is relevant to the field of posttraumatic growth and spirituality.

Introduction was useful and good leading to the research question

Methods:

-  when was the study conducted, what is the time / Duration of the Study? And how were the mothers recruited? It is not somewhat clear!

Response: Thank you very much for your valuable comment. We have added missing information in the text (in section: 2.1 Participants, p. 3)

- In the manuscript was not mentioned how long had been child cancer? how long had the mothers experienced the children's cancer?, the discussion of this point will be an enrichment for the topic and the Temporal relationship between posttraumatic growth and spirituality

Response: Thank you very much for your valuable comment. Information on the time since diagnosis is presented in Table 1 (p. 3) - almost 86% of children were diagnosed with cancer within the last 18 months. For this reason, we did not decide to analyze the subgroups (as there would be a very large disproportion in the number of respondents assigned to each subgroup). We agree that this may be an important factor and we indicated it in section: 4.1 Limitations and Future Directions, p. 7.

-              Is there ethical approval of the Study?

Response: We have added missing information in the text (in section: 2.1 Participants, p. 3). The research project was consulted with the management of both hospitals and obtained their approval.

-              Are there exclusion criteria?

Response: We did not use criteria other than those mentioned in the text.

Measures the questionnaire tolls are explained in general good.      

A suggestion for the to the Title, so it may refer to the results like:

a positive association between posttraumatic growth and spirituality in mothers of children with pediatric cancer

Response: Thank you very much for your valuable suggestion. Due to the moderate strength of the described relationships (which we write about in the section: Discussion), we decided to keep the previous title.

Reviewer 4 Report

The issue of experienced severe stress in parents of children diagnosed with cancer continues to be of interest for many researchers. This manuscript's strengths include a review of literature which is focused on the variables used in this research. Also, the sample of mothers was well chosen because they are the ones who mostly bear the burden of coping with stress and giving support to their sick children, especially when they are young. Coping with traumatic life events such as the threat of a child's terminal illness can have positive consequences (e.g., posttraumatic growth). A strong mother can more easily give strength to her child struggling with illness. Therefore, the choice of the research problem is well justified.

However, there are also some shortcomings:

  1. Introduction. The manuscript contains too many references. It would be necessary to check which references are actually needed to support the research problem posed. Could the authors select the most essential papers instead of citing all of them? Currently, there is a clear imbalance between the length of the article and the number of references.
  2. Participants
    1. Did the authors collect information about participants' religiosity and religious practices? If so, it would be helpful to provide this data. Pargament, Desai, and McConnel's (2006) findings suggest that some people become more religious due to experienced trauma and that it is related to religious involvement before the trauma. If such data was not collected, it would be appropriate to mention that in the limitations of the study.
    2. Can the authors describe the study procedure in more detail? How did participants consent to participate in the research project (written or verbal informed consent)?
    3. In Table 1, N should be capitalized because it refers to the number of total study participants.
  3. Measures. Section 2 (Materials and Methods) lacks a formally separate subsection 2.2. Measures (or Instruments), although the measurement tools are correctly described on page 4 (lines 106-137). Subsection 2.2 should be added.
  4. Results
    1. Unclear word "5umer" in lines 158-159. Please check.
    2. To enrich the analyses the authors might consider a comparative analysis between the group of mothers who scored high on the PTGI questionnaire (at least 7 sten) and those who scored low or average (see lines 152-154). Such analysis could track different dimensions of spirituality in those mothers who had noticeable posttraumatic growth. I leave it to the authors' decision whether this analysis will yield additional interesting results.
  5. Discussion
    1. The authors report positive relationships between posttraumatic growth in mothers of children with cancer and their spirituality, especially its two components: ethical sensitivity and harmony. No significant association was found between mothers' posttraumatic growth (total score) and their religious attitudes. The question is, why? It would be worthwhile to interpret this result.
  6. References
    1. References must be numbered in order of appearance in the text (including citations in tables and legends) and listed accordingly at the end of the manuscript. In the current version of the manuscript, the literature is arranged alphabetically making it difficult to follow the text.
    2. The authors should limit the number of references.

Author Response

Reviewer 4

Comments and Suggestions for Authors

The issue of experienced severe stress in parents of children diagnosed with cancer continues to be of interest for many researchers. This manuscript's strengths include a review of literature which is focused on the variables used in this research. Also, the sample of mothers was well chosen because they are the ones who mostly bear the burden of coping with stress and giving support to their sick children, especially when they are young. Coping with traumatic life events such as the threat of a child's terminal illness can have positive consequences (e.g., posttraumatic growth). A strong mother can more easily give strength to her child struggling with illness. Therefore, the choice of the research problem is well justified.

However, there are also some shortcomings:

  1. The manuscript contains too many references. It would be necessary to check which references are actually needed to support the research problem posed. Could the authors select the most essential papers instead of citing all of them? Currently, there is a clear imbalance between the length of the article and the number of references.

Response: Thank you very much for your valuable comment. As suggested, we made a selection of references (there are currently 43 items on the list).

  1. Participants
    1. Did the authors collect information about participants' religiosity and religious practices? If so, it would be helpful to provide this data. Pargament, Desai, and McConnel's (2006) findings suggest that some people become more religious due to experienced trauma and that it is related to religious involvement before the trauma. If such data was not collected, it would be appropriate to mention that in the limitations of the study.

Response: Thank you very much for your valuable comment. We agree that this may be an important factor, unfortunately, during our research, we did not collect such data, and therefore we indicated it as a limitation of our research (in section: 4.1 Limitations and Future Directions, p.7).

  1. Can the authors describe the study procedure in more detail? How did participants consent to participate in the research project (written or verbal informed consent)?

Response: Thank you very much for your valuable comment. We have added missing information in the text (in section: 2.1 Participants, p. 3)

  1. In Table 1, Nshould be capitalized because it refers to the number of total study participants.

Response: The table was corrected.

  1. Section 2 (Materials and Methods) lacks a formally separate subsection 2.2. Measures (or Instruments), although the measurement tools are correctly described on page 4 (lines 106-137). Subsection 2.2 should be added.

Response: Subsection was added.

  1. Results
    1. Unclear word "5umer" in lines 158-159. Please check.

Response: It was a typo. We have made a correction.

  1. To enrich the analyses the authors might consider a comparative analysis between the group of mothers who scored high on the PTGI questionnaire (at least 7 sten) and those who scored low or average (see lines 152-154). Such analysis could track different dimensions of spirituality in those mothers who had noticeable posttraumatic growth. I leave it to the authors' decision whether this analysis will yield additional interesting results.

Response: Thank you very much for your valuable comment. The results of the additional analysis are presented in Table 4 (p. 6).

  1. Discussion
    1. The authors report positive relationships between posttraumatic growth in mothers of children with cancer and their spirituality, especially its two components: ethical sensitivity and harmony. No significant association was found between mothers' posttraumatic growth (total score) and their religious attitudes. The question is, why? It would be worthwhile to interpret this result.

Response: Thank you very much for your valuable comment. We took into account the hypotheses regarding the lack of such a relationship in the discussion (p. 7).

  1. References
    1. References must be numbered in order of appearance in the text (including citations in tables and legends) and listed accordingly at the end of the manuscript. In the current version of the manuscript, the literature is arranged alphabetically making it difficult to follow the text.
    2. The authors should limit the number of references.

Response: The number of references has been reduced and the numbering has been improved to make it more readable.